# Parameter Identification of Lithium Battery Model Based on Chaotic Quantum Sparrow Search Algorithm

Jing Hou [1], Xin Wang [1], Yanping Su [2], Yan Yang [1,*] and Tian Gao [1]

[1] School of Electronics and Information, Northwestern Polytechnical University, Xi'an 710129, China; jhou0825@nwpu.edu.cn (J.H.); wxx2020@mail.nwpu.edu.cn (X.W.); tiangao@nwpu.edu.cn (T.G.)
[2] Tongchuan Vocational and Technical School, Tongchuan 727031, China; yanpingsu0513@163.com
[*] Correspondence: yangyan7003@nwpu.edu.cn

**Abstract:** An accurate battery model is of great importance for battery state estimation. This study considers the parameter identification of a fractional-order model (FOM) of the battery, which can more realistically describe the reaction process of the cell and provide more precise predictions. Firstly, an improved sparrow search algorithm combined with the Tent chaotic mapping, quantum behavior strategy and Gaussian variation is proposed to regulate the early population quality, enhance its global search ability and avoid trapping into local optima. The effectiveness and superiority are verified by comparing the proposed chaotic quantum sparrow search algorithm (CQSSA) with the particle swarm optimization (PSO), genetic algorithm (GA), grey wolf optimization algorithm (GWO), Dingo optimization algorithm (DOA) and sparrow search algorithm (SSA) on benchmark functions. Secondly, the parameters of the FOM battery model are identified using six algorithms under the hybrid pulse power characterization (HPPC) test. Compared with SSA, CQSSA has 4.3%, 5.9% and 11.5% improvement in mean absolute error (MAE), root mean square error (RMSE) and maximum absolute error (MaAE), respectively. Furthermore, these parameters are used in the pulsed discharge test (PULSE) and urban dynamometer driving schedule (UDDS) test to verify the adaptability of the proposed algorithm. Simulation results show that the model parameters identified by the CQSSA algorithm perform well in terms of the MAE, RMSE and MaAE of the terminal voltages under all three different tests, demonstrating the high accuracy and good adaptability of the proposed algorithm.

**Keywords:** battery; fractional-order model; parameter identification; sparrow search algorithm; chaotic mapping; quantum behavior

## 1. Introduction

Lithium-ion batteries have become the most promising energy solution by virtue of their high energy density, long life cycle and low self-discharge rate [1]. However, they are extremely intolerant of over-charging and over-discharging and prone to fire and even explosion in the case of poor monitoring. Clearly, it is essential to monitor the current, voltage and temperature and estimate the state of charge (*SOC*) and the state of health (SOH) so as to ensure battery safety and efficiency [2]. However, the direct measurement of the *SOC* is impractical; it usually needs to be estimated based on observable data such as current and voltage. The most commonly used *SOC* estimation method is the model-based method, which is usually implemented in two steps: firstly establishing a model for the battery and then using an adaptive filter, such as the Kalman filter and its variants [3,4], to estimate the *SOC*. Consequently, the battery model is the premise of *SOC* estimation. Therefore, it is requisite to develop an appropriate model which can precisely describe the dynamic process of the battery and reflect the relationship between the *SOC* and the observable data so as to improve *SOC* estimation performance.

To date, considerable research efforts have been made on battery models, which can be in general divided into three main categories: the black-box model, electrochemical

model and equivalent circuit model (ECM). Firstly, the black-box model usually employs artificial neural networks [5] or support vector machines [6] to learn the nonlinear relationship between the battery input and output. Chemali et al. [7] introduced a method employing a recurrent neural network with long short-term memory (LSTM-RNN) to perform correct *SOC* estimation for the lithium-ion battery. Chehade et al. [8] combined the advantages of LSTM and the Gaussian process and achieved accurate estimates. With a large amount of data, this method is able to estimate *SOC* without using any physical models or filters. However, since the model parameters have no explicit physical meaning, they require a large amount of data under different experimental conditions to train the model, which puts a heavy demand on the quality of the sample data as well as the computation capability [9,10]. The electrochemical model needs to develop a detailed and complex model [11,12] to describe the internal chemical reaction. It involves a variety of chemical parameters, which requires a great amount of tedious and expensive chemical experiments. As a result, this model cannot realize real-time online detection. Compared with the electrochemical model, the ECMs have been widely applied, which can mimic the electric behavior of the battery through simple circuit elements, including a serial resistor, one or more resistor-capacitor (RC) circuits and an ideal voltage source. Therefore, only a few parameters are required in ECMs. By appropriately adjusting the model structure, it is easy to reach a balance between accuracy and complexity. Hu et al. [13] compared 12 commonly used ECMs in terms of accuracy, robustness and complexity and concluded that the first-order RC model is the most suitable one. Furthermore, as the RC order increases, the model accuracy improves, whereas the computational efficiency decreases.

Related research [14,15] shows that the diffusion effect in lithium-ion batteries is more appropriate to be described based on fractional order. This fact facilitated the rapid development of the fractional-order model (FOM) of the battery. One improvement of the ECMs was to replace the ideal capacitor with a constant phase element (CPE) in the first-order RC model to more accurately simulate the behaviors of double layers on the electrode [16]. It was found that the FOM with one CPE is equivalent to an integer-order model (IOM) which has five RC networks [17]. To further improve the accuracy, a Warburg component (W) was added in series with the polarization resistor [18] or the internal ohmic resistor [19] to describe the battery dynamic characteristics more properly. Moreover, Wang et al. [20] proposed an FOM with two CPEs, which exhibited more robustness to uncertainties.

In addition to the accurate modeling of the battery, another key challenge is model parameter identification. In general, precise model parameters are not only crucial for *SOC* estimation but they can also reflect the battery SOH which is usually defined by the internal resistance or the capacity. For the IOMs of ECM, a typical method is offline identification based on the HPPC test which, however, has poor accuracy under dynamic operating conditions. Therefore, plenty of online parameter identification algorithms have been put forward, such as the recursive least squares (RLS) [21], RLS with forgetting factors (FFRLS) [22], extended Kalman filters (EKF) [23] and universal adaptive stabilizers (UAS) [24]. Moreover, some researchers have designed an online parameter identification method considering multiple time scales according to the fast and slow change characteristics of the model parameters [25]. Nevertheless, these methods are not adequate for FOM models. A heuristic optimization algorithm is one feasible solution. For example, the genetic algorithm (GA) was adopted to obtain the order of the fractional element and other model parameters [26–28]. Moreover, Su et al. [29] used the particle swarm optimization (PSO) algorithm to identify the parameters of a fractional-order two-RC circuit model. Zhang et al. [30] proposed a time-frequency-domain-based fractional-order model parameter identification method using a genetic particle swarm optimization algorithm. However, the naive forms of these algorithms are prone to fall into local optimal solutions. To account for this weakness, quantum behavior has been combined with intelligent algorithms, such as the PSO [31], artificial bee colony algorithm [32] and bacterial foraging algorithm [33].

Recently, to further improve the solution accuracy, a sparrow search algorithm (SSA) was proposed [34]. However, like PSO and GA, it may also be trapped into local optimum although the global search ability is excellent. In addition, the SSA converges slowly because a random wandering strategy is adopted when there are no neighboring peers around an individual sparrow. To address these issues, many studies have been carried out. Zhang et al. [35] proposed an improved SSA which applied the logistic chaotic mapping and adaptive parameters to the sparrow position initialization and update and obtained good configuration parameters. Zhu et al. [36] added an adaptive learning rate to accelerate the process and avoid the inefficient random wandering. Furthermore, levy flight strategies [37,38] and Gaussian variation strategies [39] have been applied to improve the ability of avoiding getting stuck in a local optimal solution.

In this study, we adopt a fractional-order model with two CPEs as a battery model and formulate the model parameter identification problem as an optimization problem. The chaotic mapping and quantum behavior strategies are combined with the SSA to minimize the sum of squared errors (SSE) between the measured and estimated voltages of the model. Experiments are performed under three different operating conditions, namely, the hybrid pulsed power characteristic test (HPPC), pulsed discharge test (PULSE) and urban dynamometer driving schedule test (UDDS). Compared with PSO, GA, GWO [40], DOA [41] and SSA, the proposed chaotic quantum sparrow search algorithm (CQSSA) exhibits superior performance in accuracy and adaptability.

The main contributions of this study are as follows:

(1) A novel chaotic quantum sparrow search algorithm (CQSSA) is proposed, which uses the quantum behavior strategy to improve the intelligence of the algorithm and the ability to jump out of the local optimum and adopts Gaussian variation to enhance the population diversity. Moreover, it employs the Tent chaotic mapping to initialize the sparrow population to improve the diversity of the initial population and accelerate the convergence rate.

(2) The proposed CQSSA algorithm is applied to realize the parameter identification of the fractional-order model of the lithium-ion battery based on the HPPC experimental data. Simulation results indicate that CQSSA can identify the model parameters much more accurately than the GA, PSO, GWO, DOA and SSA algorithms.

(3) These parameters are used in the pulsed discharge test and UDDS test to verify the adaptability of the CQSSA algorithm. Simulation results show that the parameters obtained by CQSSA also perform best under the pulsed discharge test and UDDS test, illustrating the good adaptability of the proposed algorithm under different operating conditions.

The remainder is organized as follows. Section 2 introduces the sparrow search algorithm. A series of improvements is proposed in Section 3, and the performance verification is made using six test functions. In Section 4, the experimental results and analysis for battery parameter identification are presented. The last section provides the conclusion.

## 2. Lithium-Ion Battery Model

### 2.1. Battery Modeling

Compared with the integer-order model (IOM) of ECM, the fractional-order model (FOM) can simulate more electrochemical processes of the cell. Therefore, the FOM model can realistically describe the reaction process of the cell [42,43] and give more precise predictions. To make a trade-off between accuracy and complexity, an FOM with two CPEs is constructed in this study, as shown in Figure 1.

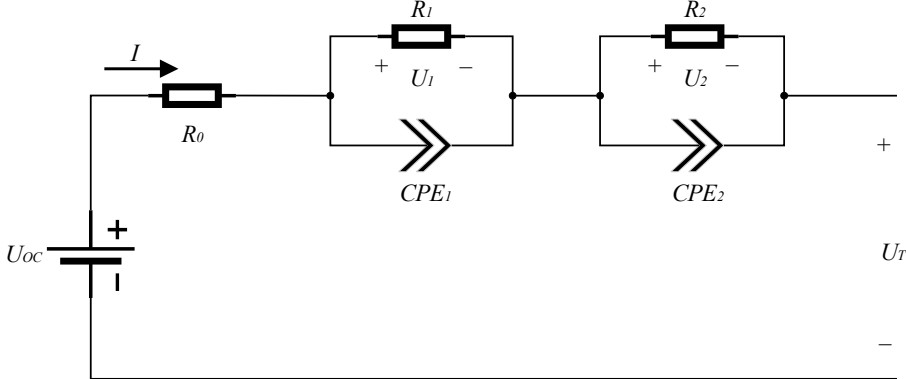

**Figure 1.** The fractional order model with two CPEs of the lithium-ion battery.

The structure is similar to that of the IOM, except for the use of the constant phase element. $R_0$ is the ohmic internal resistance; $U_{oc}$ represents the open-circuit voltage (OCV); $U_T$ denotes the terminal voltage; $I$ is the operating current; the first loop formed by $R_1$ and the constant phase element $CPE_1$ represents the lithium-ion diffusion behavior in solid phases; the second loop formed by $R_2$ and the constant phase element $CPE_2$ mimics the double layer effect. $U_1$ and $U_2$ are the voltages of the loops. $C_1$ and $C_2$ are the abbreviations of $CPE_1$ and $CPE_2$. In order to make fractional order theory applicable to Kalman filtering, the Grünwald–Letnikov (GL) definition is employed to build the fractional-order battery model. As such, we have:

$$\rho D_t^\theta f(t) = \lim_{T_S \to 0} T_S^{-\theta} \sum_{i=0}^{[(t-\rho)/T_s]} (-1)^i \binom{\theta}{i} f(t - iT_s) \tag{1}$$

where $\rho D_t^\alpha$ represents the fractional calculus operator, $\theta$ is the fractional order, $\rho$ and $t$ stand for the upper and lower bounds of the integral and $T_s$ indicates the step size.

Discretizing (1), we can obtain

$$D^\theta f(k) = T_s^{-\theta} \sum_{i=0}^{L_m} (-1)^i \binom{\theta}{i} f(k - i) \tag{2}$$

where $L_m$ is a memory length. In theory, $L_m$ should be the integer part of $t/T_s$, including every sample point. However, the larger the value of $L_m$, the greater the computational burden and the slower the computation. Therefore, we take a suitable integer for the value of $L_m$ for simplicity.

Based on this discrete expression and the mathematical description of the electrical behaviors of the battery using the FOM model

$$D^m U_1(t) = -\frac{1}{R_1 C_1} U_1(t) + \frac{1}{C_1} I(t)$$
$$D^n U_2(t) = -\frac{1}{R_2 C_2} U_2(t) + \frac{1}{C_2} I(t) \tag{3}$$

the following equations can be obtained:

$$T_s^{-m} \sum_{i=0}^{L_m} (-1)^i \binom{m}{i} U_1(k+1-i) = -\frac{1}{R_1 C_1} U_1(k) + \frac{1}{C_1} I(k)$$
$$T_s^{-n} \sum_{i=0}^{L_m} (-1)^i \binom{n}{i} U_2(k+1-i) = -\frac{1}{R_2 C_2} U_2(k) + \frac{1}{C_2} I(k) \tag{4}$$

The state of charge (*SOC*) is obtained by using the ampere-time integration method as:

$$SOC(t) = SOC_0 - \frac{\int \lambda I(t) dt}{Q_n} \tag{5}$$

where $\lambda$ is the coulombic efficiency and $Q_n$ is the nominal capacity of the battery.

The terminal voltage is given by

$$U_b(k) = U_{oc}(SOC) - U_1(k) - U_2(k) - R_0 I(k) \qquad (6)$$

where $U_{oc}(SOC)$ represents the relationship between the open-circuit voltage (OCV) and the $SOC$, which is fitted using a polynomial function based on the experimental data, shown as:

$$U_{oc} = \begin{aligned} &-152.94 \times \hat{SOC}8 + 595.53 \times \hat{SOC}7 - 931.73 \times \hat{SOC}6 + 751.62 \times \hat{SOC}5 \\ &-338.93 \times \hat{SOC}4 + 92.04 \times \hat{SOC}3 - 17.73 \times \hat{SOC}2 + 3.14 \times \hat{SOC}1 + 3.23 \end{aligned} \qquad (7)$$

Then, if we take $U_1$, $U_2$ and $SOC$ as the state vector $x$ and the battery terminal voltage as the output and current as the input, the matrix from of the state space equations can be expressed as follows:

$$\begin{cases} x_{k+1} = T_s^\theta A x_k + T_s^\theta B I_k + T_s^\theta \omega_k - \sum_{i=1}^{k+1} (-1)^i \phi_i x_{k+1-i} \\ y_k = C x_k + D I_K + U_{OC} + v_k \end{cases} \qquad (8)$$

where

$$A = \begin{bmatrix} -T_s/(R_1 C_1) & & \\ & -T_s/(R_2 C_2) & \\ & & 1 \end{bmatrix}$$

$$B = \begin{bmatrix} T_s/C_1 & T_s/C_2 & -T_s/Q_N \end{bmatrix}^T$$

$$x = \begin{bmatrix} U_1 & U_2 & SOC \end{bmatrix}^T$$

$$C = \begin{bmatrix} -1 & -1 & 0 \end{bmatrix}$$

$$D = \begin{bmatrix} -R_0 \end{bmatrix}$$

$$y = \begin{bmatrix} U_b \end{bmatrix}$$

$$\theta = \begin{bmatrix} m & n & 1 \end{bmatrix}^T$$

and $\phi_i$ represents the generalized binomial coefficient, which is calculated as follows:

$$\phi_i = \begin{pmatrix} \theta \\ i \end{pmatrix} = diag\left[ \begin{pmatrix} m \\ i \end{pmatrix} \begin{pmatrix} n \\ i \end{pmatrix} \begin{pmatrix} 1 \\ i \end{pmatrix} \right]$$

$$\begin{pmatrix} r \\ i \end{pmatrix} = \begin{cases} 1 & i = 0 \\ r(r-1)\ldots(r-i+1)/i! & i > 0 \end{cases}$$

### 2.2. Objective Function of Parameter Identification

For an FOM model, the traditional online identification algorithms such as RLS are not suitable owing to the complexity brought by fractional order calculus. Intelligent offline optimization algorithms are usually effective for this problem. The FOM model parameters include ohmic internal resistance ($R_0$), polarized internal resistance ($R_1$, $R_2$), a constant phase element ($C_1$, $C_2$) and two orders of fractional calculus ($m$, $n$) in this study. They can be determined by minimizing the sum of squared errors (SSE) between the actual measured terminal voltage $V_r$ and the estimated terminal voltage $V_e$ of the battery model. Therefore, the objective function of optimization is established as:

$$F(x) = \min \sum_{i=1}^{T} \left( V_e{}^i - V_r{}^i \right)^2 \qquad (9)$$

where $T$ indicates the total number of voltage sampling points.

### 3. Sparrow Search Optimization Algorithm

By studying the group predation of the sparrow species, Xue [34] proposed a sparrow search algorithm (SSA) which abstracts the flock predation behavior of the sparrows into a producer-follower model.

The basic idea of the SSA algorithm is as follows. It divides the individuals in the sparrow population into two categories, producers and followers, which take on different tasks in the predation behavior. Usually, producers are responsible for identifying and discovering abundant food sources and providing feeding directions for followers. The follower remains in the same direction as the producer with the best position, and some followers monitor the positions of the producers and compete with other followers for food to improve their own predation rates. The ratio of the producers to followers remains constant throughout the sparrow population, but each sparrow has the potential to become a producer depending on its fitness value.

Following the behavior of the sparrows in predation and escape from predators, a mathematical model is established. Assuming there exists a population of sparrows with population size $N$, the position $X$ of the sparrow population is represented as:

$$
X = \begin{bmatrix}
X_{1,1} & X_{1,2} & \cdots & \cdots & X_{1,D} \\
X_{2,1} & X_{2,2} & \cdots & \cdots & X_{2,D} \\
\vdots & \vdots & \vdots & \vdots & \vdots \\
X_{N,1} & X_{N,2} & \cdots & \cdots & X_{N,D}
\end{bmatrix}
\tag{10}
$$

where each row of the matrix represents the position of a sparrow. $X_{i,j}$ indicates the position of the $j$th ($j = 1,2, \ldots ,D$) dimension for the $i$th ($i = 1,2, \ldots ,N$) sparrow. Each individual sparrow represents a feasible solution.

The producers, which generally occupy 10–20% of the entire sparrow population, need to search for food with full freedom in the search space and lead the entire population to forage. Therefore, the producers play a crucial role in the guarantee of global optimization performance. Their positions are updated as follows:

$$
X_{i,j}^{t+1} = \begin{cases}
X_{i,j}^t \cdot \exp\left(\frac{-i}{\alpha \cdot iter_{\max}}\right) & ALV < ST \\
X_{i,j}^t + Q \cdot L & ALV \geq ST
\end{cases}
\tag{11}
$$

where $t$ denotes the current iteration number, $iter_{\max}$ denotes the maximum iteration number, $\alpha$ represents a random number uniformly distributed between (0, 1], $Q$ is also a random number that follows the standard normal distribution and $L$ stands for an all-one matrix. $ST$ denotes the safety threshold and $ALV$ denotes the alert value. An alert value less than the safety threshold means that there is no predator near this location, and the producers can conduct an extensive search following this search direction. On the contrary, the producers need to lead the followers to other safe areas.

The follower positions are updated as follows:

$$
X_{i,j}^{t+1} = \begin{cases}
Q \cdot \exp\left(\frac{X_{worst}^t - X_{i,j}^t}{i^2}\right) & i > N/2 \\
X_P^{t+1} + \left| X_{i,j}^t - X_P^{t+1} \right| \cdot A^+ \cdot L & otherwise
\end{cases}
\tag{12}
$$

where $X_{worst}^t$ and $X_P^t$ denote the global worst position and best position of the producer. $A$ is a matrix of size $1 \times D$ whose elements are randomly assigned with 1 or $-1$ and $A^+$ indicates the additive inverse, which can be defined as $A^+ = A^T(AA^T)^{-1}$. When $i > N/2$, this follower has a low fitness value. It is largely hungry and may compete and find food more actively, while the rest of the followers are monitoring the position of the producer and competing for food.

In addition, 10–20% of sparrows in the population take charge of early warning. When they become aware of a dangerous situation, the locations of these sparrows are updated as:

$$X_{i,j}^{t+1} = \begin{cases} X_{best}^t + \beta \cdot \left| X_{i,j}^t - X_{best}^t \right| & f_i > f_g \\ X_{i,j}^t + K \cdot \left( \dfrac{\left| X_{i,j}^t - X_{worst}^t \right|}{(f_i - f_w) + \varepsilon} \right) & f_i = f_g \end{cases} \tag{13}$$

where $X_{best}^t$ denotes the global best position, $\beta$ and $K$ represent the step control parameters of the direction of movement, which can be any number between $[-1, 1]$, $\varepsilon$ represents a small positive constant used to avoid dividing by zero, $f_i$ is the fitness value of sparrow $i$, $f_g$ denotes the current optimal fitness value and $f_w$ denotes the current worst fitness value. If $f_i > f_g$, the current sparrow will move toward the position of the producer when it realizes the danger. If they are equal, it means the current sparrow is already at the middle of the population and it will approach the other sparrows when it realizes the danger.

## 4. Improvements on the Sparrow Search Algorithm

To enhance the intelligence of the population evolution and improve the overall performance of the algorithm, a chaotic quantum sparrow search algorithm (CQSSA) is proposed in this study. It adopts the Tent chaos mapping to expand the population diversity and uses the quantum behavior and Gaussian variation strategy to intelligentize the motion trajectory of the population's individuals and improve global search capability while effectively avoiding getting trapped in a local optimal solution.

### 4.1. Quantum Behavior Improvement Strategies

In recent years, quantum mechanics has made great progress in both theoretical research and engineering applications. Many studies have been inspired by the behavior and properties of quantum, including algorithm design [44,45] and circuit quantum electrodynamics system design [46]. In [47,48], the quantum behavior improvement strategy inspired by the Delta potential well model in quantum mechanics was adopted to improve the PSO algorithm to extend the search range of the particles. It uses the bound states to describe the aggregation of particles in quantum space, in which the particles can appear at any space point with a certain probability density and can be searched throughout the feasible solution space without dispersion to infinity. As a result, combining the quantum behavioral improvement strategy with SSA can increase the possibilities of the foraging behavior of the sparrows and enhance the intelligence of the group search behavior.

Mathematically, based on quantum Delta potential well theory, the iterative update of the individual sparrow position can be obtained as:

$$X_{i,j}^{t+1} = p_{i,j}^t \pm \frac{L_{i,j}^t}{2} \ln(1/u_{i,j}^t) \tag{14}$$

where $i$ denotes the $i$th individual, $j$ denotes the $j$th dimension of the solution and $t$ represents the time instant. $p_{i,j}{}^t$ denotes the center of the identified potential well, to which the individual converges in probability. $L_{i,j}{}^t$ denotes the characteristic length of the Delta potential well. $u_{i,j}{}^t$ is a random variable uniformly distributed on $[0, 1]$.

First, the characteristic length of the Delta potential well ($L_{i,j}{}^t$) is evaluated as:

$$L_{i,j}^t = 2\alpha \cdot \left| mbest_j^t - X_{i,j}^t \right| \tag{15}$$

where $mbest_j^t$ is the average best position of the individual and $\alpha$ is the contraction-expansion coefficient of the current iteration. An efficient method to set $\alpha$ is to decrease the value of $\alpha$ linearly during the search [49]. That is,

$$\alpha = (\alpha_{\max} - \alpha_{\min}) \frac{iter_{\max} - t}{iter_{\max}} + \alpha_{\min} \tag{16}$$

$$mbest_j^t = \frac{1}{N} \sum_{i=1}^{N} pbest_{i,j}^t \tag{17}$$

where $\alpha_{\max}$ and $\alpha_{\min}$ are the boundary values of the shrinkage-dilation coefficient, which generally take on values of 1 and 0.5, and $pbest_{i,j}^t$ denotes the optimal position of the individual from the beginning to the current iteration.

Next, the center of the identified potential well $p_{i,j}^t$ is expressed as a random point of attraction between the global optimal position $gbest_j^t$ and the individual optimal position $pbest_{i,j}^t$ and obtained as:

$$p_{i,j}^t = \varphi_j^t \cdot pbest_{i,j}^t + (1 - \varphi_j^t) \cdot gbest_j^t \tag{18}$$

where $\varphi_j^t$ is a random variable between [0, 1].

Finally, substituting (15) and (18) into (14), the final sparrow position is updated as

$$X_{i,j}^{t+1} = \varphi_j^t \cdot pbest_{i,j}^t + (1 - \varphi_j^t) \cdot gbest_j^t \pm \alpha \cdot \left| mbest_j^t - X_{i,j}^t \right| \cdot \ln(1/u_{i,j}^t) \tag{19}$$

### 4.2. Tent Chaotic Mapping

Due to the merit of the unification of randomness and ergodicity, the Tent chaotic mapping [50,51] can enhance the population variability during the initialization. Therefore, to improve the population quality, the Tent chaotic mapping is applied to initialize the SSA algorithm.

The process of sparrow population initialization using the Tent chaotic mapping method is as follows:

**Step 1**: Generate the initial value of chaos $z_0^k$.

**Step 2**: Adding a random number to the Tent mapping expression to eliminate the effect of small periodic points [52], $N$ $k$-dimensional chaotic variables can be generated as

$$z_{i+1} = \begin{cases} z_i/0.5 + rand(0,1) \times \frac{1}{N}, 0 \leq z_i \leq 0.5 \\ (1 - z_i)/0.5 + rand(0,1) \times \frac{1}{N}, 0.5 < z_i \leq 1 \end{cases} \tag{20}$$

where $N$ is the number of sparrows and rand (0, 1) is a random number uniformly distributed in the range [0, 1].

**Step 3**: Map the chaotic variables to the solution space so as to achieve the population initialization using the inverse mapping of the generated chaotic variables, which is

$$X_k = L_k + (H_k - L_k) \cdot z_k \tag{21}$$

where $L_k$ and $H_k$ are the lower and upper bounds of the values of the individual sparrow position, respectively.

### 4.3. Chaotic Quantum Sparrow Search Algorithm

Combing the above improvement strategies with SSA, the CQSSA algorithm is proposed. The basic idea is as follows. Firstly, the Tent chaotic mapping is used for initialization to generate a more diverse sparrow population to improve the iteration efficiency and enhance the global search capability. Secondly, the Gaussian variation is performed to produce more types of individuals and thus increase the population diversity if the individual optimal solution is smaller than the individual average optimal solution. Moreover,

the quantum behavior is employed, which can search throughout the solution space and reduce the probability of getting trapped in the local optimum.

The processing steps are as follows:

**Step 1**: Initialize the number of iterations $iter_{max}$, the number of sparrows in the population $N$ and the dimension of the solution space $D$ and determine the upper bounds $U$ and lower bounds $L$ of the parameters.

**Step 2**: Use the Tent chaos mapping to generate a chaotic sequence and map them to the solution space using an inverse mapping, and then initialize the positions of the sparrow population by (20) and (21).

**Step 3**: According to the cell model and the objective function (9), calculate the fitness values of sparrows $Fit_p$ and compare them with the individual optimal position $X_p$, global optimal position $X_{best}$, global worst position $X_{worst}$ as well as the optimal fitness value $Fit_{best}$. Then, rank the sparrows according to the fitness value to get better and worse positions among the sparrows.

**Step 4**: Regard the top 20% of the ranked sparrows as producers. Use (11) to update their positions.

**Step 5**: Take sparrows other than the producers as followers. Moreover, use (12) to update the positions.

**Step 6**: Select 20% of sparrows randomly as guard sparrows, whose positions are updated by (13).

**Step 7**: Calculate the average fitness value of the population $Fit_{Ave}$ and compare it with the optimal individual fitness value $Fit_p$.

If $Fit_p$ is less than $Fit_{Ave}$, regard the individual within the population and update its position to enhance the population diversity by using Gaussian variation. Gaussian variation applies a random number that obeys a Gaussian distribution to perturb the original position. The Gaussian variation equation is shown as follows:

$$X_{Gaussian} = X \cdot (1 + N(m, \sigma^2)) \tag{22}$$

where $X_{Gaussian}$ is the position after Gaussian variation, $X$ is the original position and $N(m, \sigma^2)$ is a Gaussian distributed random variable.

If $Fit_p$ is greater than $Fit_{Ave}$, regard the individual as far away from the community and use the quantum behavior by (19), which draws the individual towards a certain position between the individual optimal and global optimal positions.

Then, compare the updated position with the original position and select the best individual for the next iteration.

**Step 8**: After completing an iteration, calculate the fitness values and update the optimal position $X_p$, global optimal position $X_{best}$, global worst position $X_{worst}$ and global optimal fitness value $Fit_{best}$ of the sparrow individual.

**Step 9**: Determine whether the current number of iterations reaches the maximum number of iterations. If so, the algorithm ends and the output result is returned; otherwise, return to Step 3.

The algorithmic flow of the CQSSA algorithm is shown in Figure 2.

### 4.4. Simulation and Verification

Six benchmark test functions including unimodal and multimodal functions are used to verify the effectiveness of CQSSA by comparing with PSO, GA, GWO, DOA and SSA. This study uses an Intel®Core™i5 processor and MATLAB 2018a for algorithm verification on Windows 10. The selected test functions are shown in Table 1.

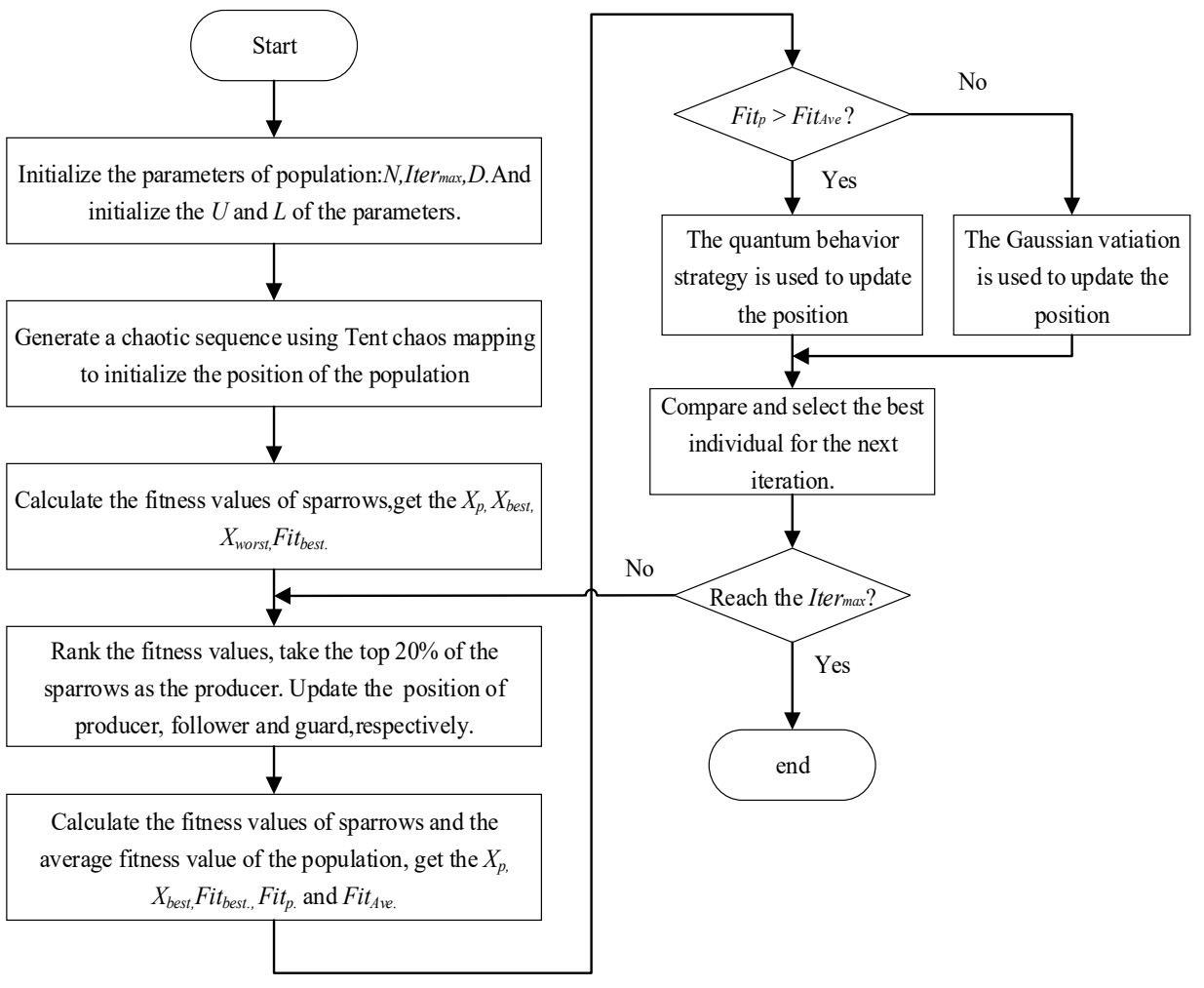

**Figure 2.** The algorithmic flow of the CQSSA algorithm.

**Table 1.** Test Function (Dim = 30).

| Name | Function | Dim | Initial Range | $F_{min}$ |
|---|---|---|---|---|
| Quaritic | $F1 = \sum\limits_{i=1}^{n} i x_i^4 + random[0,1)$ | 30 | $[-1.28, 1.28]$ | 0 |
| Schwefe l2.26 | $F2 = \sum\limits_{i=1}^{n} -x_i \sin(\sqrt{\lvert x_i \rvert})$ | 30 | $[-500, 500]$ | $-418.9829 \times \dim$ |
| Rosenbrock | $F3 = \sum\limits_{i=1}^{n-1} [100(x_{i+1} - x_i^2)^2 + (x_i - 1)^2]$ | 30 | $[-30, 30]$ | 0 |
| Griwank | $F4 = \frac{1}{4000} \sum\limits_{i=1}^{n} x_i^2 - \prod\limits_{i=1}^{n} \cos(\frac{x_i}{\sqrt{i}}) + 1$ | 30 | $[-600, 600]$ | 0 |
| Ackley | $F5 = -20\exp(-0.2\sqrt{\frac{1}{n}\sum\limits_{i=1}^{n} x_i^2}) - \exp\left(\frac{1}{n}\sum\limits_{i=1}^{n}\cos(2\pi x_i)\right)$ $+ 20 + e$ | 30 | $[-32, 32]$ | 0 |
| Penalized1 | $F6 = \frac{\pi}{n}\left\{10\sin(\pi y_1) + \sum\limits_{i=1}^{n-1}(y_i - 1)^2[1 + 10\sin^2(\pi y_i + 1)] + (y_n - 1)^2\right\}$ $+ \sum\limits_{i=1}^{n} u(x_i, 10, 100, 4)$ $y_i = 1 + \frac{x_i+1}{4}$ $u(x_i, a, k, m) = \begin{cases} k(x_i - a)^m & x_i > a \\ 0 & -a < x_i < a \\ k(-x_i - a)^m & x_i < -a \end{cases}$ | 30 | $[-50, 50]$ | 0 |

F1 and F3 are unimodal functions which can reflect the convergence and exploration abilities of the algorithm. F2, F4, F5 and F6 are multimodal functions which have multiple

local extreme points which can reflect the local search and global search capabilities of the algorithm.

To ensure the verification accuracy, six algorithms are implemented independently 30 times on each function. The population size is 100. For the PSO algorithm, the learning factors and the inertia coefficient are set as $c_1 = c_2 = 1.49618$ and $w = 0.7298$. For the GA algorithm, the genes are encoded in binary and the roulette method is used to select the genes that enter the next generation. The solution of each dimension is represented by 6 genes, the chromosome length is $6 \times$ Dim, and the mutation and crossover probabilities are set to 0.05 and 0.5, respectively. For GWO, the weight of the location distance of the wolf pack decreases linearly and $r_1$ and $r_2$ are random vectors within [0, 1]. For DOA, $\beta_1$ is a scaling factor that can change the trajectory of the dingoes and it is a random number uniformly distributed over [−2, 2]; $\beta_2$ is a random number uniformly generated in the interval [−1, 1]. For SSA, the proportions of producers and guards are both 20%. For CQSSA, the maximum and minimum quantum contraction expansion coefficients are 1.0 and 0.5, respectively. The performance of the six different algorithms under each test function is shown in Table 2.

**Table 2.** The simulation results of the six algorithms on different test functions.

| Function | | PSO | GA | GWO | DOA | SSA | CQSSA |
|---|---|---|---|---|---|---|---|
| F1 | Best | $1.6792 \times 10^{-1}$ | $8.0814 \times 10^{-5}$ | $6.2782 \times 10^{-5}$ | $4.7453 \times 10^{-6}$ | $9.2096 \times 10^{-6}$ | $\mathbf{4.2640 \times 10^{-6}}$ |
| | Ave | $7.1141 \times 10^{-1}$ | $1.3028 \times 10^{-3}$ | $4.8361 \times 10^{-4}$ | $1.0664 \times 10^{-4}$ | $8.1352 \times 10^{-5}$ | $\mathbf{7.5527 \times 10^{-5}}$ |
| | Std | $3.0420 \times 10^{-1}$ | $1.3417 \times 10^{-3}$ | $6.0031 \times 10^{-4}$ | $1.3815 \times 10^{-4}$ | $7.6052 \times 10^{-5}$ | $\mathbf{5.3268 \times 10^{-5}}$ |
| F2 | Best | $-9.29 \times 10^{3}$ | $\mathbf{-1.25 \times 10^{4}}$ | $-1.25 \times 10^{4}$ | $-8.39 \times 10^{3}$ | $-9.61 \times 10^{3}$ | $-1.04 \times 10^{4}$ |
| | Ave | $-7.31 \times 10^{3}$ | $\mathbf{-1.24 \times 10^{4}}$ | $-1.19 \times 10^{4}$ | $-6.53 \times 10^{3}$ | $-8.67 \times 10^{3}$ | $-9.47 \times 10^{3}$ |
| | Std | $8.78 \times 10^{2}$ | $\mathbf{2.64 \times 10^{2}}$ | $7.71 \times 10^{2}$ | $9.58 \times 10^{2}$ | $5.08 \times 10^{2}$ | $4.65 \times 10^{2}$ |
| F3 | Best | $1.4081 \times 10^{2}$ | $3.6877 \times 10^{2}$ | $2.5251 \times 10^{1}$ | $2.8700 \times 10^{1}$ | $2.4401 \times 10^{-10}$ | $\mathbf{9.1537 \times 10^{-11}}$ |
| | Ave | $6.0121 \times 10^{2}$ | $1.3177 \times 10^{3}$ | $2.5889 \times 10^{1}$ | $2.8829 \times 10^{1}$ | $4.3553 \times 10^{-6}$ | $\mathbf{5.9915 \times 10^{-7}}$ |
| | Std | $4.3758 \times 10^{2}$ | $3.0390 \times 10^{3}$ | $1.8254 \times 10^{-1}$ | $4.5647 \times 10^{-2}$ | $1.7396 \times 10^{-5}$ | $\mathbf{1.5278 \times 10^{-6}}$ |
| F4 | Best | $6.3826 \times 10^{-1}$ | $1.6803 \times 10^{0}$ | 0.0000 | 0.0000 | 0.0000 | **0.0000** |
| | Ave | $1.1170 \times 10^{0}$ | $1.7286 \times 10^{0}$ | $2.1608 \times 10^{-3}$ | 0.0000 | 0.0000 | **0.0000** |
| | Std | $1.4760 \times 10^{-1}$ | $2.0729 \times 10^{-1}$ | $5.6434 \times 10^{-3}$ | 0.0000 | 0.0000 | **0.0000** |
| F5 | Best | $3.9496 \times 10^{0}$ | $4.2819 \times 10^{0}$ | $8.8818 \times 10^{-16}$ | $8.8818 \times 10^{-16}$ | $8.8818 \times 10^{-16}$ | $\mathbf{8.8818 \times 10^{-16}}$ |
| | Ave | $5.3482 \times 10^{0}$ | $4.4290 \times 10^{0}$ | $3.9672 \times 10^{-15}$ | $1.4803 \times 10^{-15}$ | $8.8818 \times 10^{-16}$ | $\mathbf{8.8818 \times 10^{-16}}$ |
| | Std | $1.0122 \times 10^{0}$ | $3.3094 \times 10^{-1}$ | $2.0298 \times 10^{-15}$ | $1.3467 \times 10^{-15}$ | 0.0000 | **0.0000** |
| F6 | Best | $1.3308 \times 10^{0}$ | $5.3099 \times 10^{-1}$ | $1.9654 \times 10^{-5}$ | $4.4535 \times 10^{-2}$ | $7.1302 \times 10^{-19}$ | $\mathbf{1.5705 \times 10^{-32}}$ |
| | Ave | $6.3947 \times 10^{0}$ | $1.3249 \times 10^{0}$ | $2.7880 \times 10^{-4}$ | $1.9846 \times 10^{-1}$ | $2.7284 \times 10^{-14}$ | $\mathbf{1.5705 \times 10^{-32}}$ |
| | Std | $3.6420 \times 10^{0}$ | $1.2239 \times 10^{0}$ | $1.2421 \times 10^{-3}$ | $1.3124 \times 10^{-1}$ | $5.0740 \times 10^{-14}$ | $\mathbf{5.5674 \times 10^{-48}}$ |

For the unimodal function F1, PSO performs worst on this high-dimensional problem, whereas CQSSA has significantly better accuracy and stability than the other five algorithms. For function F2, GA has the best optimization result, followed by CQSSA, and they both significantly outperform the other algorithms. For function F3, CQSSA has the best optimization result. Furthermore, it can be clearly seen that the optimization result of CQSSA is significantly superior to the other algorithms for the multimodal functions F4–F6. It indicates that the proposed CQSSA algorithm performs better in convergence and stability probably due to the good capability of escaping from the local extreme.

The convergence characteristic curves of the six algorithms under the six benchmark functions are shown in Figure 3. The convergence criterion is the minimum number of iterations to reach the required accuracy or the highest convergence accuracy under the fixed iteration number. Overall, the CQSSA algorithm converges faster and has higher convergence accuracy. Specifically, CQSSA achieves the best value in about 200 iterations in F1, which is much smaller than the other algorithms. For F2, CQSSA also converges to a good fitness value, second only to the GA algorithm. For F3 and F4, the convergence results of PSO, GA, GWO and DOA are comparable and both worse than SSA and CQSSA. Moreover, CQSSA has faster convergence speed and higher convergence accuracy than SSA. For F5, both SSA and CQSSA achieve the best convergence accuracy and GWO and DOA are slightly inferior, while PSO and GA perform the worst. For F6, it can be seen from the convergence curve that although GWO and DOA have the best fitness values at the 200th

iteration, they fall into local optimal solutions after that. However, CQSSA finally achieves the best convergence accuracy. Different than F3 and F6, the convergence processes of the CQSSA and SSA are almost the same under F4 and F5. The best convergence results are observed for both. In general, compared to SSA, CQSSA uses an improved strategy to make a more intelligent search and jump out of the local optimal solutions in time. This allows CQSSA to exhibit good convergence accuracy on both unimodal and multimodal test functions in high dimensions.

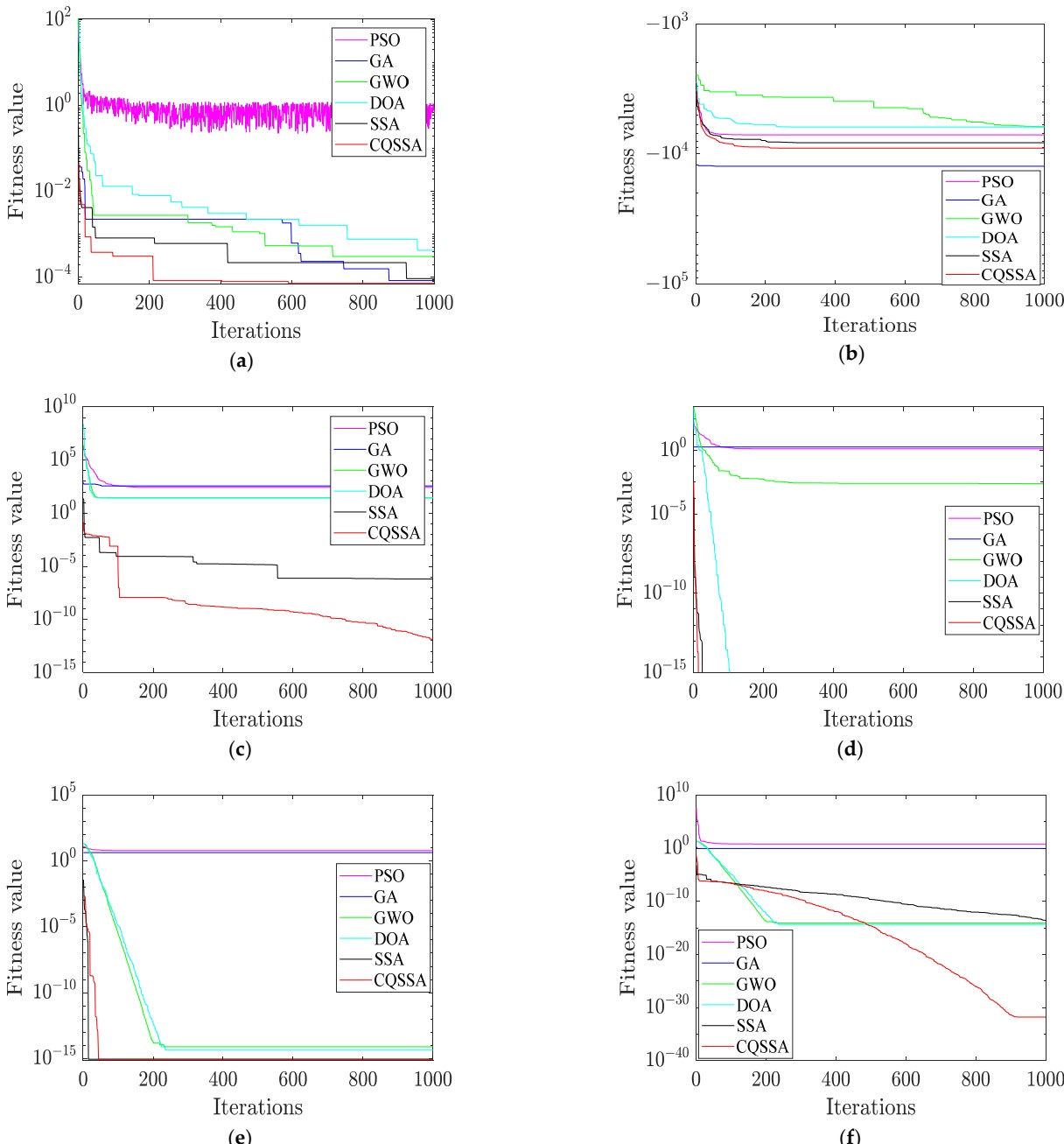

**Figure 3.** The convergence characteristics of the six algorithms under six test functions. (**a**) The convergence curve of algorithms on F1. (**b**) The convergence curve of algorithms on F2. (**c**) The convergence curve of algorithms on F3. (**d**) The convergence curve of algorithms on F4. (**e**) The convergence curve of algorithms on F5. (**f**) The convergence curve of algorithms on F6.

## 5. Battery Model Parameter Identification Based on Improved Sparrow Search Optimization Algorithm

### 5.1. Experiment Platform

The ICR18650 lithium-ion battery has been widely used in grid power storge, electric vehicles and consumer electronics. Therefore, it is used to perform the simulated discharge tests on a battery tester (NEWARE CT-4008T-5V6A). The battery specification is shown in Table 3. Three different discharge tests, including the HPPC test [53], pulsed discharge test (PULSE) and UDDS test, are performed. By using the test data under the three different tests, the performance of CQSSA is verified in the parameter identification of the cell model.

**Table 3.** The battery specification description.

| Battery Type | Nominal Capacity | Nominal Voltage | Upper Cut-Off Voltage | Lower Cut-Off Voltage |
|---|---|---|---|---|
| ICR18650-26F | 2600 mAh | 3.7 V | 4.2 V | 2.75 V |

In one cycle of the HPPC test, the battery is firstly charged for 20 s and rested for 80 s, then discharged for 20 s and rested for 80 s and finally stood for 1 h after 10% *SOC* is consumed. The pulse discharge test is to conduct a discharge for 10 min with a current of 0.5 C and rest for 20 min, repeating this cycle until the cut-off voltage of the discharge is reached.

The UDDS test is a test procedure used by the U.S. Environmental Protection Agency (EPA) in 1972 to certify vehicle emissions. It is commonly referred to as the "urban test" in terms of application scenarios since it usually represents the urban driving conditions for light-duty vehicles. Moreover, it includes a wide range of driving behaviors with varying intensities. The current in this study is scaled down according to the tolerance of the battery, as shown in Figure 4.

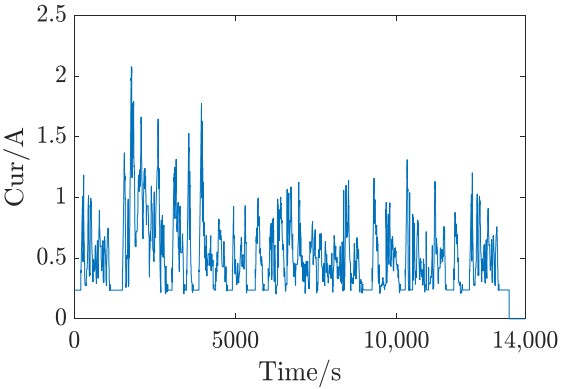

**Figure 4.** The current waveform under UDDS test.

### 5.2. Experimental Results

Firstly, six different optimization algorithms (CQSSA, SSA, DOA, GWO, PSO and GA) are respectively utilized to identify the parameters of the FOM model based on the HPPC test data. Then, the identified parameters are employed to obtain the estimated terminal voltages in the PULSE and UDDS tests. By comparing the errors between the estimated and measured terminal voltages, the effectiveness and superiority of the CQSSA algorithm for cell model parameter identification are verified.

The value ranges of the identified model parameters are shown in Table 4. The parameters of the PSO, GA, GWO, DOA, SSA and CQSSA algorithms have been set in accordance with Section 4.4.

**Table 4.** The value range of model parameters.

|  | $R_0/\Omega$ | $R_1/\Omega$ | $R_2/\Omega$ | $C_1/F$ | $C_2/F$ | $n_1$ | $n_2$ |
|---|---|---|---|---|---|---|---|
| Minimum | 0.01 | 0.001 | 0.001 | 800 | 10,000 | 0 | 0 |
| Maximum | 0.5 | 0.05 | 0.05 | 50,000 | 500,000 | 1 | 1 |

### 5.2.1. Parameter Identification Results under HPPC Test

Table 5 shows the battery parameters identified by six different algorithms under the HPPC test. Table 6 shows the estimation errors of the terminal voltage with six different algorithms. The MAE, RMSE, MaAE and SSE are used to reflect the accuracy of the cell model. Clearly, in terms of SSE, MAE, RMSE and MaAE, CQSSA has the best accuracy while GA has the worst accuracy. Compared with SSA, CQSSA has 4.3%, 5.9% and 11.5% improvement in MAE and MaAE, respectively. Figure 5 shows the graphical comparison of terminal voltages. It can be seen that the CQSSA has the smallest error, which is consistent with the above analysis. In addition, we observed that the maximum absolute error usually occurs when the input current has a sudden change. Therefore, the smallest MAE and MaAE states that the model parameters identified by CQSSA not only perform optimally in the average error but also maintain a smaller error in the current loading and unloading stage.

**Table 5.** Model parameter identification results of six optimization algorithms under HPPC test.

|  | PSO | GA | GWO | DOA | SSA | CQSSA |
|---|---|---|---|---|---|---|
| $R_0/\Omega$ | 0.1516 | 0.1501 | 0.1486 | 0.1508 | 0.1493 | 0.1470 |
| $R_1/\Omega$ | 0.0493 | 0.0329 | 0.0321 | 0.0353 | 0.0487 | 0.0316 |
| $R_2/\Omega$ | 0.0423 | 0.0298 | 0.0201 | 0.0381 | 0.0479 | 0.0207 |
| $C_1/F$ | 7341.34 | 6933.34 | 5603.23 | 8299.90 | 5085.70 | 3542.91 |
| $C_2/F$ | 10,040.81 | 367,778.78 | 315,286.39 | 121,562.45 | 10,040.81 | 465,848.78 |
| $n_1$ | 0.9196 | 0.9714 | 0.9470 | 0.9397 | 0.7520 | 0.9201 |
| $n_2$ | 0.9861 | 0.4571 | 0.6952 | 0.9397 | 0.9134 | 0.9959 |

**Table 6.** Terminal voltage errors of six different algorithms under HPPC test.

|  | PSO | GA | GWO | DOA | SSA | CQSSA |
|---|---|---|---|---|---|---|
| $SSE/V^2$ | 0.4022 | 0.5378 | 0.4337 | 0.4359 | 0.4138 | **0.3666** |
| MAE/mV | 2.5611 | 2.8209 | 2.7038 | 2.6342 | 2.6521 | **2.5355** |
| RMSE/mV | 3.5573 | 4.1136 | 3.6943 | 3.7037 | 3.6085 | **3.3966** |
| MaAE/mV | 21.9365 | 19.5735 | 17.8127 | 20.6484 | 18.9053 | **16.7271** |

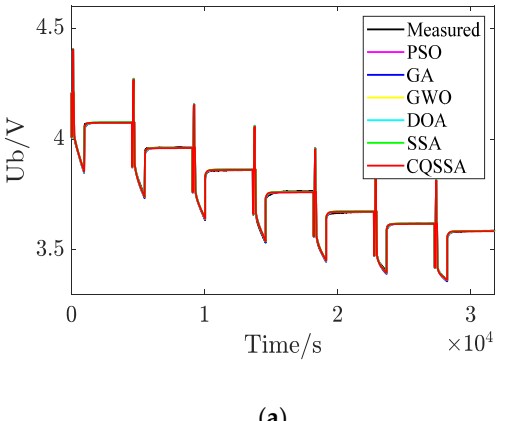
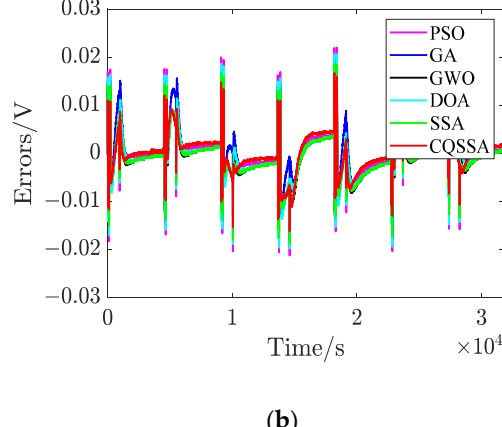

(**a**)　　　　　　　　　　　　　　　　　　　　　　　　(**b**)

**Figure 5.** The comparison of estimated and measured voltages under HPPC test. (**a**) The comparison of measured and estimated terminal voltages. (**b**) The comparison of terminal voltage errors of six algorithms.

Figure 6 shows the iterative processes of the six algorithms. Obviously, the fitness values (SSE) all gradually converge. However, CQSSA converges to under 0.4 $V^2$ with the least number of iterations. Furthermore, the final fitness value is also the smallest, revealing the highest convergence accuracy.

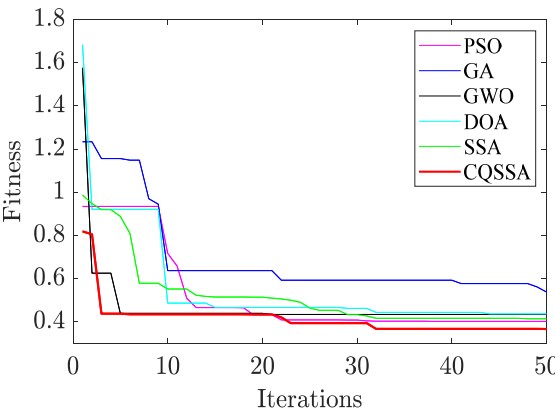

**Figure 6.** The convergence characteristics of six algorithms under HPPC test.

5.2.2. Model Verification under PULSE Test

The experimental data in the pulsed discharge test are used to verify the model accuracy. Furthermore, the terminal voltages are obtained using the identified parameters given in Table 5. Figure 7 presents the comparison of the estimated and measured terminal voltages. Moreover, the errors of terminal voltage with six different groups of FOM parameters under the PULSE test are displayed in Table 7. It is clear that the MAE, RMSE and MaAE with the battery parameters identified by CQSSA are 3.3242 mV, 4.3880 mV and 17.6197 mV, respectively, which are all the smallest among the six algorithms. Moreover, although the FOM parameters obtained by PSO are not far behind CQSSA in MAE and RMSE, they have the worst performance in terms of MaAE. These findings reveal the superior performance of the proposed algorithm in the total error.

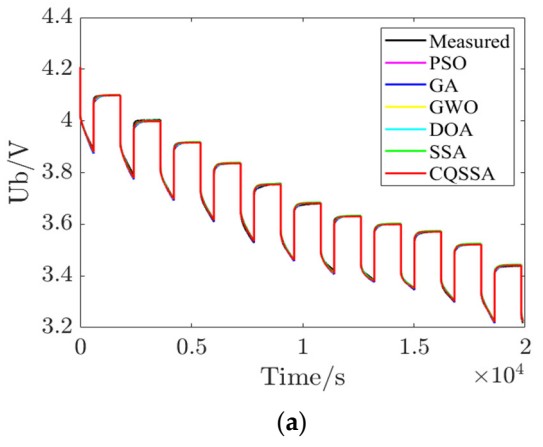

(**a**)

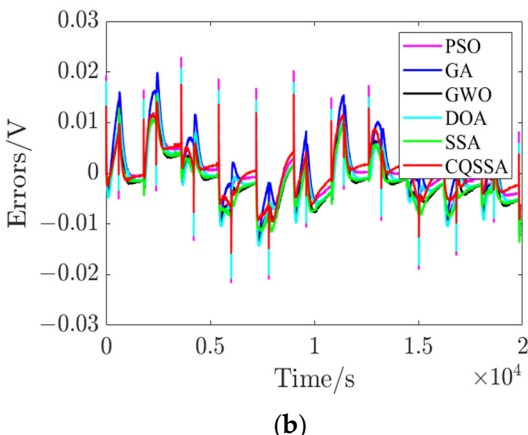

(**b**)

**Figure 7.** The comparison of estimated and measured voltages under PULSE test. (**a**) The comparison of measured and estimated terminal voltages. (**b**) The comparison of terminal voltage errors of six algorithms.

**Table 7.** Terminal voltage errors with six different groups of FOM parameters under PULSE test.

|  | PSO | GA | GWO | DOA | SSA | CQSSA |
|---|---|---|---|---|---|---|
| MAE/mV | 3.5833 | 4.1330 | 4.2167 | 3.9922 | 4.0579 | **3.3242** |
| RMSE/mV | 4.7421 | 5.5249 | 5.1398 | 5.1015 | 4.9662 | **4.3880** |
| MaAE/mV | 22.9509 | 19.9808 | 17.9134 | 20.8257 | 18.8050 | **17.6197** |

### 5.2.3. Model Verification under UDDS Test

Table 8 shows the terminal voltage errors with six different groups of FOM parameters under the UDDS test. It is clear that the CQSSA outperforms the other algorithms in terms of MAE and RMSE of the terminal voltage although both GWO and DOA have smaller MaAE than CQSSA. In addition, the MAE, RMSE and MaAE of the proposed algorithm are larger than these in the HPPC test and PULSE test. This phenomenon can also be seen in the estimated and actual terminal voltages shown in Figure 8. The overall terminal voltage errors fluctuate between 60 mV and 100 mV due to the large current variation during the discharge process in UDDS test.

**Table 8.** Terminal voltage errors with six different groups of FOM parameters under UDDS test.

|         | PSO     | GA      | GWO        | DOA     | SSA      | CQSSA      |
|---------|---------|---------|------------|---------|----------|------------|
| MAE/V   | 6.8785  | 6.7375  | 7.8769     | 7.4498  | 7.8005   | **6.5561** |
| RMSE/V  | 9.3589  | 9.3657  | 10.2224    | 9.8503  | 10.1413  | **9.0941** |
| MaAE/mV | 92.0834 | 93.0041 | **90.0701**| 90.7653 | 91.7841  | 91.5958    |

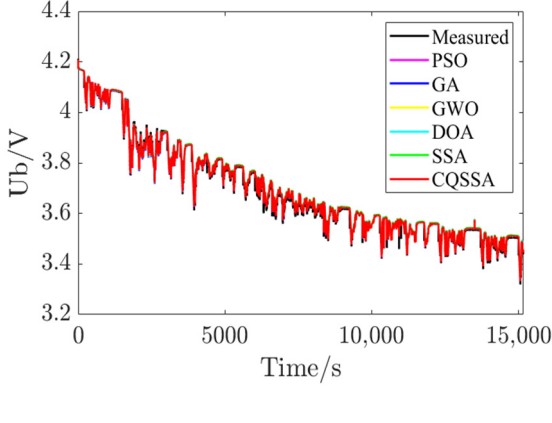

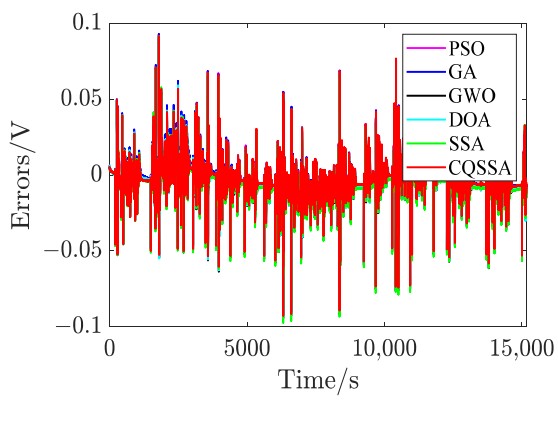

(**a**)  (**b**)

**Figure 8.** The comparison of estimated and measured voltages under UDDS test. (**a**) The comparison of measured and estimated terminal voltages. (**b**) The comparison of terminal voltage errors of six algorithms.

### 5.2.4. Discussion

From Table 6, we can see that using the model parameters identified by the proposed CQSSA algorithm can achieve the smallest MAE, RMSE and MaAE under the HPPC test. It implies that the battery model identified by CQSSA agrees better with the actual battery. Furthermore, it can be seen that the parameters obtained by CQSSA also perform best under the pulsed discharge test and UDDS test from Tables 7 and 8. It shows the CQSSA-based battery model has better adaptability to different operating conditions.

For the convergence characteristics of the algorithms shown in Figure 6, CQSSA has a better solution at the beginning of the iteration since it uses a Tent chaotic mapping to improve population diversity. Furthermore, the improvements of Gaussian variation mixed with quantum behavior enables it to overstep the local optima and obtain better convergence accuracy than PSO, GA, GWO, DOA and SSA.

To sum up, the advantages and disadvantages of CQSSA algorithm can be analyzed by the performance of CQSSA and other five algorithms in benchmark functions and practical applications. The main advantage of the proposed algorithm is the high optimization accuracy and the ability of jump out of local optimal solutions due to combination with the Tent chaotic mapping and quantum behavior strategy. Especially when solving problems with multiple local optimal solutions, the superiority of the CQSSA algorithm becomes more prominent. Moreover, the proposed algorithm has a clear structure and few parameters. It

is easy to apply in practical engineering to solve different optimization problems. However, the complexity of the algorithm is relatively higher than that of the other algorithms.

## 6. Conclusions

In this study, a novel chaotic quantum sparrow search algorithm (CQSSA) is proposed which combines Tent chaotic mapping, quantum strategy and Gaussian variation to improve the abilities of global search and escaping from local optimal solution. Six benchmark test functions with single peaks and multiple peaks are selected to validate the great convergence performance of the CQSSA by comparing with PSO, GA, GWO, DOA and SSA. Then, the six algorithms are applied to the parameter identification of the second-order FOM model of the lithium-ion battery based on the HPPC experimental data. Finally, these parameters are used in the pulsed discharge test and UDDS test to verify the adaptability of the CQSSA algorithm. Simulation results indicate that CQSSA can identify the model parameters much more accurately than the other three algorithms based on the HPPC test. Furthermore, the parameters obtained by CQSSA also perform best under the pulsed discharge test and UDDS test, illustrating the good adaptability of the proposed algorithm under different operating conditions. This study provides an improved method for battery model parameter identification, which is of significance for ensuring the precision of the cell model and the accurate *SOC* and SOH estimation.

**Author Contributions:** Conceptualization, X.W. and T.G.; Data curation, J.H., Y.Y. and T.G.; Formal analysis, J.H. and Y.Y.; Funding acquisition, J.H. and Y.Y.; Investigation, Y.S. and T.G.; Methodology, J.H. and X.W.; Project administration, Y.S.; Resources, J.H., Y.Y. and T.G.; Software, X.W.; Supervision, J.H., Y.S. and Y.Y.; Validation, J.H., X.W. and T.G.; Visualization, X.W. and Y.S.; Writing—original draft, X.W.; Writing—review & editing, J.H., Y.S. and Y.Y. All authors have read and agreed to the published version of the manuscript.

**Funding:** This work was supported in part by the National Natural Science Foundation of China under grant 52007156.

**Institutional Review Board Statement:** Not applicable.

**Informed Consent Statement:** Not applicable.

**Data Availability Statement:** The data that support the findings of this study are available from the corresponding author upon reasonable request.

**Conflicts of Interest:** The authors declare no conflict of interest.

## Abbreviations

| | |
|---|---|
| *SOC* | the state of charge |
| ECM | equivalent circuit model |
| LSTM-RNN | recurrent neural network with long short-term memory |
| RC | resistor-capacitor |
| FOM | fractional-order model |
| IOM | integer order model |
| W | Warburg component |
| HPPC | hybrid pulse power characterization |
| PULSE | pulsed discharge test |
| UDDS | urban dynamometer driving schedule |
| RLS | recursive least squares |
| FFRLS | recursive least squares with forgetting factors |
| EKF | extended Kalman filter |
| UAS | universal adaptive stabilizer |
| GA | genetic algorithm |
| PSO | particle swarm optimization algorithm |

| GWO | grey wolf optimization algorithm |
| DOA | dingo optimization algorithm |
| SSA | sparrow search algorithm |
| CQSSA | chaotic quantum sparrow search algorithm |
| SSE | sum of squared errors |
| MAE | mean absolute error |
| RMSE | root mean square error |
| MaAE | maximum absolute error |
| GL | Grünwald–Letnikov definition |
| $U_{oc}(SOC)$ | the relationship between the open-circuit voltage and the SOC |
| A | system matrix |
| B | control matrix |
| C | observation matrix |
| D | transition matrix |
| $F(x)$ | objective function |

**Parameters**

| $U_{oc}$ | open-circuit voltage |
| $R_0$ | ohmic internal resistance |
| $R_1, R_2$ | polarization resistance |
| $CPE_x\ C_x$ | constant phase element |
| $U_1\ U_2$ | voltages of the loops |
| $U_T$ | the terminal voltage |
| $D^*$ | fractional calculus operator |
| $T_s$ | the step size |
| $\theta$ | the fractional order |
| $L_m$ | memory length |
| $m\ n$ | the fractional order of two CPEs |
| $k$ | discrete moments |
| $\lambda$ | coulombic efficiency |
| $Q_n$ | nominal capacity of the battery |
| $x$ | the state vector |
| $\varphi_i$ | generalized binomial coefficient |
| $V_e$ | estimated terminal voltage |
| $V_r$ | actual measured terminal voltage |
| $T$ | total number of voltage sampling points |
| $X$ | the positions of sparrow population |
| $N$ | sparrow population size |
| $D$ | dimension of the solution space |
| $ALV$ | alert value |
| $ST$ | safety threshold |
| $X_{i,j}{}^t$ | the position of the $j$th dimension for the $i$th sparrow in the $t$th iteration |
| $X_{worst}{}^t$ | global worst position |
| $X_{best}{}^t$ | global best position |
| $X_P{}^t$ | the best position of the producer |
| $A$ | a matrix of size $1 \times D$ |
| $\beta, K$ | step control parameters of the direction |
| $\varepsilon$ | a small positive constant used to avoid dividing by zero |
| $f_i$ | fitness value of sparrow $i$ |
| $f_g$ | the current optimal fitness value |
| $f_w$ | the current worst fitness value |

| | |
|---|---|
| $p_{i,j}{}^t$ | the center of the identified potential well |
| $L_{i,j}{}^t$ | characteristic length of the Delta potential well |
| $u_{i,j}{}^t$ | random variable uniformly distributed on [0, 1] |
| $\alpha$, $\alpha_{\max}$, $\alpha_{\min}$ | contraction-expansion coefficient, max = 1, min = 0.5 |
| $mbest_j{}^t$ | the average best position of the individual |
| $pbest_j{}^t$ | the optimal position of the individual |
| $gbest_j{}^t$ | the global optimal position |
| $\varphi_j^t$ | a random variable between [0, 1] |
| $z^k$ | chaos initialization value |
| $L_k$, $H_k$ | the lower and upper bounds |
| $iter_{\max}$ | the maximum number of iterations |
| $Fit_p$ | the optimal individual fitness value |
| $Fit_{Ave}$ | the average fitness value of the population |
| $X_{Gaussian}$ | the position of sparrow after Gaussian variation |
| $Fit_{best}$ | the global optimal fitness value |

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
