# Peer review of "Parameter Identification of Lithium Battery Model Based on Chaotic Quantum Sparrow Search Algorithm"

_applsci, doi:10.3390/app12147332_

Round 1
Reviewer 1 Report
The manuscript considers a very interesting and relevant research problem, parameter Identification of Lithium Battery Model Based on Chaotic Quantum Sparrow Search Algorithm. This considered research topic is within the scope of the journal. The paper is well organized. Figures are mostly well constructed and informative.
Provide a table with relevant nomenclature. There are certain typographical issues that need to be fixed.
e.g. 1.” Their positions are as follows: “ 185. Should be ” Their positions are updated as follows:
The paper could benefit if the overview provide a brief overview or the sparrow search algorithm (SSA) to delving into detail of the algorithm. For example, the paper should introduce different types of sparrows, such as producers and scroungers and their respective roles. Without an adequate overview of the sparrow search algorithm (SSA), this section is difficult to follow.
In addition, the readability of the manuscript would improve if the same notation used is the same as in the original paper where SSA is proposed. For example, in this manuscript ALV is used to denote the “alert” value, while in the original paper, R2 is used to denote “alarm” value.
Define A+ = AT ( AAT )− 1
Make sure that all parameters are described. For example, parameter fw does not seems to be explained.
Also ensure that parameters are precisely described. For example, in the manuscript eps is denoted as “a non-zero value”, but it should be denoted as a small positive constant used to avoid dividing by zero.
In equation (14) there is a +- sign. In [37], the sparrow position is more precisely defined, se equation (4) i [37]. The sign depends on a specific u value. Please explain this.
Provide the reference for the chaotic tent map.
Provide a reference for the equation (16) describing how contraction-expansion coefficient is evaluated.
Provide a reference for the equation (20) and provide a more detailed explanation of this equation,
A more extensive comparison with more recently proposed metaheuristic algorithms is required to adequately evaluate the proposed algorithm. Additional experiments are required.
Reviewer 2 Report
In this paper, the authors discussed a Chaotic Quantum Sparrow Search Algorithm Six benchmark test functions with single-peak and multi-peak are selected to validate the great convergence performance of the CQSSA by comparing with PSO, GA and SSA. Then, the four algorithms are applied to the parameter identification of the second-order FOM model of the lithium-ion battery based on the HPPC experimental data. It is shown that this study provides an improved method for the battery model parameters identification, which is of significance for ensuring the precision of the cell model and the accurate SOC and SOH estimation.
This paper contains some new and interesting results after the following are taken into account
1) The English language should be improved
2) In the figure caption "Figure 1 The fractional order model of the lithium-ion battery" more information have to be added
3) In page 4, equation 4 should be fixed
4) In section "Chaotic Quantum Sparrow Search Algorithm" some new references may be added i.e.
M. Zidan, S. Aldulaimi, H. Eleuch, Analysis of the Quantum Algorithm based on Entanglement Measure for Classifying Boolean Multivariate Function into Novel Hidden Classes: Revisited, Appl. Math. Inf. Sci. Volume 15 > No. 5 (2021) PP: 643-647: doi:10.18576/amis/150513
Khalida Inayat Noor, Muhammad Aslam Noor, Hamdy M. Mohamed, Quantum Approach to Starlike Functions, Appl. Math. Inf. Sci. Volume 15, No. 4 (2021) PP: 437-441 doi:10.18576/amis/150405
Nikolai N. Bogolyubov, Jr., Andrey V. Soldatov, Time-Convolutionless Master Equation for Multi-Level Open Quantum Systems with Initial System-Environment Correlations, Appl. Math. Inf. Sci. Volume 14, No. 5 (2020) PP: 771-780 doi:10.18576/amis/140504
T. Said, A. Chouikh, M. Bennai, N Two-Transmon-Qubit Quantum Logic Gates Realized in a Circuit QED System, Appl. Math. Inf. Sci. Volume 13 > No. 5 (2019) PP: 839-846 doi:10.18576/amis/130518
T. Said, A. Chouikh, M. Bennai, Two-Step Scheme for Implementing N Two-Qubit Quantum Logic Gates Via Cavity QED, Appl. Math. Inf. Sci. Volume 12, No. 4 (2018) PP: 699-704 doi:10.18576/amis/120404
Nikolai N. Bogolyubov, Jr., Andrey V. Soldatov, Time-Convolutionless Master Equation for Multi-Level Open Quantum Systems with Initial System-Environment Correlations, Appl. Math. Inf. Sci. Volume 13 > No. 5 (2019) PP: 725-734 doi:10.18576/amis/130504
5) It is not clear why the authors considered "ICR18650 lithium battery to perform the simulated discharge tests on 368 a battery tester?
Reviewer 3 Report
The authors used the Chaotic Quantum Sparrow Search Algorithm for Parameter Identification of Lithium Battery Model. The mathematical formulation is well developed, and the results are well discussed. The paper can be accepted for publication after minor revision:
- The authors mentioned that ‘’Related research shows that the diffusion effect in lithium-ion battery is more appropriate to be described based on the fractional order’’ without referring to any published work.
- the novelty of the work is to be clearly stated.
- The advantages and limitations of the Chaotic Quantum Sparrow Search Algorithm are to be mentioned.
- what is the convergence criterion?
- more details on SOC and SOH estimation are to be added.
- the English level is to be improved.
- the discussion part is relatively short and need to be more developed.
- the introduction is to be extended by adding some recently published papers related to the subject such as:
https://doi.org/10.1016/j.est.2022.104595
https://doi.org/10.1007/s10973-020-10370-1
https://doi.org/10.1016/j.jpowsour.2021.230725
https://doi.org/10.2298/TSCI180211144A
https://doi.org/10.1016/j.ensm.2021.10.023
Round 2
Reviewer 1 Report
The authors have considered all the comments.
I believe the authors have put forward a significant effort to revise the paper.
I believe the manuscript has greatly improved.
After author explanations, I completely agree with on your take regarding “alarm” value.
Thank you for the explanations on other issues as well.
I have a couple of minor suggestions:
It might be a good idea to state that the Gaussian variation is used to enhance the population diversity in the section The main contributions of this study where you describe the novel Chaotic Quantum Sparrow Search Algorithm (CQSSA) as this is an important feature of this algorithm.
Furthermore, consider the following sentence in abstract “Firstly, an improved sparrow search algorithm combined with the Tent chaotic mapping and quan-10 tum behavior strategy is proposed to regulate the early population quality, enhance its global search 11 ability and avoid trapping into local optimum.” It might be a good idea to also mention Gaussian variation strategy as one of the main characteristics of the improved SSA algorithm.
